# Estimating the Environmental Impact of Green IoT Deployments

**DOI:** 10.3390/s23031537

**Published:** 2023-01-30

**Authors:** Edoardo Baldini, Stefano Chessa, Antonio Brogi

**Affiliations:** Department of Computer Science, University of Pisa, Largo B. Pontecorvo 3, 56127 Pisa, Italy

**Keywords:** green IoT, green computing, energy harvesting systems

## Abstract

The Internet of Things (IoT) is demonstrating its huge innovation potential, but at the same time, its spread can induce one of highest environmental impacts caused by the IoT industry. This concern has motivated the rise of a new research area aimed at devising green IoT deployments. Our work falls in this research area by contributing to addressing the problem of assessing the environmental impact of IoT deployments. Specifically, we propose a methodology based on an analytical model to assess the environmental impact of an outdoor IoT deployment powered by solar energy harvesting. The model inputs the specification of the IoT devices that constitute the deployment in terms of the battery, solar panel and electronic components, and it outputs the energy required for the entire life-cycle of the deployment and the waste generated by its disposal. Given an existing IoT deployment, the models also determine a functionally equivalent baseline green solution, which is an ideal configuration with a lower environmental impact than the original solution. We validated the proposed methodology by means of the analysis of a case study conducted over an existing IoT deployment developed within the European project RESCATAME. In particular, by means of the model, we evaluate the impact of the RESCATAME system and assess its impact with respect to its baseline. In a scenario with a 30-year lifespan, the model estimates for the system more than 3 times the energy required by its baseline green solution and a waste for a volume 15 times greater. We also show how the impact of the baseline increases when assuming deployments in locations at increasing latitudes. Finally, the article presents an implementation of the proposed methodology as a web service that is publicly available.

## 1. Introduction

The impact of the information and communication technologies industry on gas emissions is growing fast and, driven by datacenters, communication networks and smartphones, it is expected to reach around 14% of the total global greenhouse gas emissions by 2040 [1].

The Internet of Things (IoT) contribution to the global carbon footprint is not precisely estimated so far, but the growing proliferation of IoT devices will definitely create a new load on the carbon footprint. In addition, the energy needed to produce these devices could potentially dwarf the contribution of all the other traditional computing devices. An even more toxic problem is related to the E-Waste. Like many other hardware devices, IoT nodes are also made with hazardous substances and equipped with batteries that are difficult to manage and recycle. The E-Waste generated in 2016 was already 44.7 million tonnes, with expectations of a fast growth [2].

The studies on the impact on energy and E-Waste of technological information and communication industries, which fall within the area of Green Computing, highlight the urgency of improving the hardware manufacturing process and disposal and, more recently, have put its focus also on the IoT. Specifically, the green IoT aims at reducing the greenhouse effect of the IoT by acting on its entire life-cycle [3]. In turn, this is achieved by means of a holistic approach along four different fronts [4]: *Green design* (employ energy-efficient and eco-friendly components), *Green manufacturing* (minimize electricity, raw materials and hazardous wastes in the manufacturing process), *Green use* (reduce energy consumption and minimize the use of batteries to reduce the exposure to toxic substances) and *Green disposal* (recycle dismissed devices, reuse components and eliminate hazardous resources from the devices). In the last years, many solutions were proposed to reduce the environmental impact of IoT applications (e.g., [5,6,7,8]), but what is still missing are concrete measures to estimate the environmental impact of IoT applications in their entire life-cycle.

This work is precisely motivated by the lack of such measures. In particular, we focus on a specific class of IoT deployments, namely outdoor IoT deployments making use of solar-based energy harvesting for the production of energy. We propose a model for the assessment of the impact of such deployments in all the phases of production, use, maintenance and disposal in terms of the required energy (in megajoules—MJ) and in kilograms (kg) of waste due to their disposal. Given the input, the parameters of an IoT deployment and the specifications of its devices, the model also provides a *functionally equivalent baseline green solution*, which, for a given device, is defined as a device with the same hardware components (to ensure the baseline can keep the same software and functionality) but with the minimum-sized energy harvesting and battery subsystems that can sustain the operation of the device in ideal weather conditions. This baseline is an ideal configuration with a lower environmental impact than the original device, which we use as a yardstick to assess the impact of the evaluated deployment.

To illustrate the use of the proposed model, we present a case study where we assess the environmental impact of the air quality monitoring system developed by the European project RESCATAME [9]. This system exploits solar-powered sensors to acquire information and to implement efficient traffic management policies. In this case study, we also show how the system impact varies depending on the place (latitude) of its deployment. Finally, we also provide an IoT impact calculator web service, to make the model usable and to show the differences between an existing deployment and its baseline green version.

To the best of our knowledge, our work is one of the first attempts to evaluate the environmental impact of IoT devices during the production, utilization and disposal phases, also considering the impact of the energy harvesting system and maintenance. Note however that this work is also subject to some limitations. In particular, because the IoT is still an emerging technology, precise and up-to-date data relating to the energy consumed for their production and the impact of their disposal is often not available. Secondly, we limit our work to IoT devices harvesting solar energy, which are thus suitable only for outdoor deployments, and we focus only on IoT devices with energy harvesting and sensing subsystems. A detailed discussion of the limitations is reported in the conclusions.

The rest of the work is organized as follows: Section 2 presents the background of our work, including the information about the impact of the production and disposal of IoT devices, and the methods to assess the costs for the maintenance of an IoT deployment, and Section 3 reviews the state of the art. We then introduce, in Section 4, the model of the IoT deployments to which we refer in the rest of the work. In Section 5, we introduce our model to estimate the impact of the IoT deployments, and in Section 6, we present a case study based on the deployment of the RESCATAME European project. In Section 7, we introduce the tool to assess the impact of IoT deployments and, finally, in Section 8, we summarize the main contributions of the work, we analyze its limitations and we discuss the future work.

## 2. Background

This section presents the background information on which the rest of the work leverages. Specifically, we introduce information about the impact of the production and disposal of the different components of a solar energy harvesting IoT device, namely the device itself, the solar panel and the batteries (Section 2.1, Section 2.2 and Section 2.3). We also introduce the concept of predictive maintenance in Section 2.4 and, finally, we present the general architecture of the energy harvesting IoT devices in Section 2.5.

### 2.1. Production and Disposal of IoT Devices

Many technological industries use a life-cycle assessment (LCA) [10] to assess the environmental impact of their products (from production to disposal and recycling) and publish their results per product or as global reports. However, to the best of our knowledge, none of the IoT manufacturers provide data on the impact of their products. Considering this, our work aims at providing a model capable of estimating the impact on the energy and waste of IoT devices by leveraging on the assumption that, to estimate manufacturing energy, only the semiconductor elements of an IoT device are taken into account. We believe this is an acceptable solution because semiconductors are the backbone of today’s technology and are used in all types of electronics components. For the waste impact, we consider the printed circuit board of the devices (PCB). PCBs account for 3% of the global waste electric and electronic equipment (WEEE) and their impact is documented enough to estimate their percentage of non-recyclability.

The microelectronic industry is characterized by the volume of the chemical elements, gases, water and energy needed to make a square centimeter of product. The list of such gases and chemicals depends on the production process and technological evolution, and it is difficult to find reliable information about it. The works in [11,12] estimate that producing 1 cm${}^{2}$ of silicon wafer requires a variety of chemicals, from 5 to 29 liters (L) of high-purity water, between 445 and 924 g/cm${}^{2}$ of elemental gases (e.g., oxygen, nitrogen and argon) and an average energy consumption of 1.54 kWh/cm${}^{2}$. In particular, the work in [12] estimates that the inputs to generate a finite product are 630 times the mass of a microchip, thus highlighting how the semiconductor industry has a significant environmental impact.

With the rapid evolution of semiconductor devices, the number of consumers of electronic products based on a printed circuit board (PCB) has increased rapidly. This amount of electronics was translated in 2016 into 44.7 million tons of E-Waste [2]. PCB waste is a resource not adequately exploited and can contain up to 60 sundry elements, some of which are hazardous and others precious (e.g., gold, silver and copper). The work in [13] reports that the PCBs are 3% of the WEEE, and in the 27 European states, only 25% of the WEEE is collected and processed. To estimate how much a device will contribute to the global E-Waste, we consider the following material composition of a PCB and the relative percentages of recyclability [13,14]: metals for the 30% that is recoverable for the 80%, glass fiber for the 40% recyclable for 95% and epoxy resin for the 30% that seems to be not recyclable in those processes. Therefore, 38% of a PCB’s weight is considered to be landfill waste.

### 2.2. Production and Disposal of Solar Panels

The production of solar panels proceeds through several steps [15] that are energy hungry. Starting from high-purity solar-grade silicon, the production process varies according to the technology (monocrystalline and polycrystalline). An estimate of the energy spent in the silicon manufacturing process is given in [16,17]. From this study, we took the average of the reported data, obtaining 5476.1 MJ/m${}^{2}$ for monocrystalline (mono-Si) production and 4676.1 MJ/m${}^{2}$ for polycrystalline (poly-Si) manufacturing.

On the global scale, a fast increase in the waste consequent to the disposal of solar panels is expected as it is moving from 593 tons of waste in 2012 to the predicted 8,238,967 tons in 2050 [18]. A process to recover some types of solar panels already exists, and considering the forecast regarding the future waste of these panels, other processes will be defined. To estimate the percentage of waste produced by a solar panel at the end of its life, we consider the material composition of the panel and the percentage of the non-recyclability of these elements [18]. The solar panel elements are highly recoverable and the percentage of waste intended for the landfill is around 16.02% of their weight.

Solar panels are considered very reliable components. Indeed, many studies report a lifetime longer than 20 years with a fixed annual degradation rate between 0.5 and 1% [19]. This trend is mainly characterized by crystalline silicon technology.

### 2.3. Production and Disposal of Batteries

Our estimate of the environmental impact of a battery is based on the works [20,21,22,23]. In particular, to provide an estimate of the energy spent on the production of a battery, we calculate the mean of the data published in [20] for the different battery technologies, obtaining 164.8 and 204.143 MJ/kg for lithium and NiMh, respectively.

Although the metals contained in these batteries are recoverable in a good percentage, they contain other materials that cannot be reused. In addition, there is still no efficient program for the disposal of exhausted batteries. In our study, 55.2 and 67% of the lithium and NiMh weight is considered as intended for the landfill. The percentage of waste for a lithium battery was estimated considering the waste generated by the Pyrometallurgical and Hydrometallurgical recycling process [22]. The work in [23] reports that 33% of a NiMh battery can be recovered with valuable Ni content.

The processes of battery charge/discharge are not 100% efficient. In particular, the charge efficiency is defined as the ratio between the net energy of the battery obtained during charging over the energy extracted from a power source; the discharge efficiency is defined as the ratio between the energy extracted from the battery when it is discharged over the net energy charge of the battery; and the charge–discharge efficiency (also called battery efficiency), which is defined as the product of the charge and discharge efficiencies [24].

The battery lifespan is influenced by both the charge/discharge rate and the environmental temperature, and its modeling is complex. Many solutions have been proposed over the years [25,26], but a simple and analytical solution has not yet been established for the battery technologies considered. For this reason, to estimate the lifespan of a battery, we consider the cycles indicated in the datasheet, assuming one cycle per day. Because the data reported in the datasheet are valid for ideal conditions, in our model, we apply a degradation percentage of 35% that reduces the expected number of cycles. To estimate this percentage, we considered the study [27] that executes tests on the batteries with different loads and at different temperatures.

### 2.4. Maintenance

Maintenance extends the lifetime of an IoT deployment by monitoring, repairing and replacing every single part of it when necessary. In general, maintenance can be either corrective or preventive. Corrective maintenance is executed in response to a failure, while preventive maintenance operates in advance, to prevent the occurrence of failures. In our model, we assume that maintenance must guarantee the correct behavior of the devices, solar panels and batteries for the entire duration of the application. Without one of these three components, the system is destined to fail and stop running. For this reason, we assume multi-component preventive maintenance, which consists of a systematic inspection, detection, and the prevention of emerging failures. These three operations are executed at regular time intervals. Furthermore, multi-component-preventing maintenance may also employ a strategy of preventive components’ replacement such as, for example, one based on an age model [28] where each component is replaced at a given deadline. However, considering that in our case devices comprise different components with different deadlines, we need a more flexible strategy. For this reason, we assume opportunistic maintenance [28] in which it is possible to take advantage of the intervention for the replacement of a component to perform further maintenance operations also on other components. In this way, the cost of the intervention is shared between several components.

To find the optimal scheduling of maintenance operations that allows minimizing their environmental impact, we exploit 0-1 multi-objective linear programming optimization, which is a special case of linear programming where the objective variables at each time interval assume the value 0 (no operation performed) or 1 (maintenance operation performed). In the problem of multi-objective linear programming, it is not always possible to find a solution that is simultaneously the optimal solution for all the objective functions; therefore, the Pareto optimality is used. A solution is Pareto optimal (or nondominated) if and only if there do not exist other feasible solutions that simultaneously improve all objective functions. Hence, a Pareto optimal solution is a trade-off between all the objective functions that minimize all objective functions, although it may not be the optimal solution for each of them.

### 2.5. IoT Energy Harvesting Architectures

Energy harvesting contributes to extending the lifetime and efficiency of IoT devices while reducing the frequent maintenance due to the battery replacement that increases the costs and the environmental impact of an IoT application. Among all energy sources, solar energy is one of the most used as it provides the greatest power density, it is highly reliable and it supplies energy directly to the system without the need for circuit rectification [29].

The design of a solar energy harvesting system is very well documented in the literature. It consists of four basic components [30] as shown in Figure 1: a solar panel, an output regulator, a battery and the device itself. The solar panel supplies electricity to the device by exploiting the solar irradiance. The amount of energy that it can generate depends on the dimension and technology of its cells. Its power density can be estimated as $15\;\mathrm{mW}/{\mathrm{cm}}^{2}$ [29]. However, the size and the technology of the solar panel are only two of the characteristics that must be taken into account in the design phase. Another crucial factor is the installation site. The irradiance level changes according to the geographical location; countries closest to the equator have a higher level of irradiance than the countries near the poles. The output regulator modifies the output voltage of the battery to match the voltage required by the device. The battery needs to be rechargeable and the most used technologies are lithium and NiMh batteries. Different battery technologies have a different impact on their production and disposal and have different behavior once deployed. Indeed, lithium batteries have a higher power density, but the NiMh ones are less affected by the temperature [31]. A battery is characterized by at least two parameters, the maximum amount of energy it can store (capacity) and the charge/discharge efficiency. Other characteristics that may be considered are how the battery responds to changes in the temperature and in the load. Finally, the device is the component that consumes more energy. Estimating the daily energy required by the device is crucial for the correct design of the energy harvesting system. Other systems require the use of additional components, such as Buck–Boost or maximum power point tracking (MPPT). These elements can help manage the flow of the current from the solar panel to the battery, but on the other hand, they consume energy, introduce inefficiencies and can force the solar panel to work far from its maximum power point (MPP) [31].

In many applications, the components are rather oversized to deal with variable environmental conditions. In our study, we show how far a real implementation is from a baseline one. The baseline is defined as a (more ecological) solution in which each component is sized exactly to satisfy the daily load of the device in ideal environmental conditions, considering all the inefficiencies and the power loss in the entire system.

## 3. Related Work

The environmental impact of devices such as smartphones, servers and network equipment has been widely studied. For instance, ref. [1] presents a detailed and rigorous analysis of the global carbon footprint due to computing technologies, including both the production and the operational energy of hardware and devices, as well as the operational energy for the supporting communication infrastructures.

On the other hand, the study of the impact of IoT devices is still in its early stages. The work in [1] provides a model to estimate the global emission footprint of the information and communication technologies industry. It considers desktops, notebooks, monitors, smartphones, tablets, datacenters and communication networks. The authors based the study on the Life-cycle Annual Footprint parameter. This parameter considers the annual energy consumption of a device plus its production energy amortized over its useful lifetime. Thanks to this parameter and an intensive data collection process, it is possible to estimate the global footprint of a device and make predictions on future trends. However, this model does not consider all the life-cycle phases of a device and it is focused only on the energy consumption of the devices considered.

Many other studies are based on the life-cycle assessment (LCA) framework proposed in [32] to provide the environmental impact of different devices. This framework consists of four phases: (1) the goal and scope definition in which the scope of the study, the functional unit and the type of impacts must be defined, (2) a life-cycle inventory analysis in which the user must compile the inventory of the inputs and outputs of the product analyzed during its life-cycle, (3) a life-cycle assessment that aims at associating the inventory data with the environmental impact studied and (4) a life-cycle interpretation that exploits the results of the inventory and assessment phases to reach the conclusions consistent with the goal and scope of the study.

An LCA is widely used in many different fields. However, as described in [10,33,34], a proper assessment of a technological product is a complex task. These products are always in rapid evolution and the materials composition and the production processes vary quickly, making the life-cycle inventory analysis challenging. Furthermore, accessing the data of each component of a piece of hardware or of a device is a hard task. These problems can lead to variable or contradictory results.

Over the years, several models based on an LCA have been proposed with the aim to facilitate and accelerate the assessment process. In [35], the environmental impact of smartphones is assessed by defining a five-step model to identify the environmental weak points of a product. After the identification of the key life-cycle stage through a simple LCA-based method, the study proposes a set of environmental benchmarking parameters that allow the identification of the weak points of the product. Finally, the model defines a series of eco-design strategies that allow improving the weak points. Although this model can facilitate the identification of the weak points of a product, its first step requires the use of the LCA and it then suffers for the same limitations of the original LCA framework.

To accelerate the evaluation process of the original LCA, ref. [36] verifies several ways to simplify the life-cycle assessment framework. The study evaluates three main approaches: (1) excluding some environmental impact categories, (2) excluding life-cycle phases and (3) using secondary data. The authors present the results of the environmental assessment of smartphones. However, the simplified version of the LCA can lead to neglecting important information on the environmental impact and obtaining inaccurate results.

Other works address the IoT from the point of view of its energy efficiency. In particular, ref. [37] proposes two algorithms to minimize power and battery charge drains of IoT-based sensor devices during media transmission for smart cities. Ref. [38] proposes a cross-layer-based optimization algorithm for optimizing the energy efficiency and battery lifetime of sensor-embedded IoT devices collecting a large amount of data. Ref. [39] proposes two video transmission algorithms for energy-efficient video streaming through wireless micro medical devices for smart healthcare homes.

Our study instead aims at determining the environmental impact of a specific case: IoT devices equipped with a solar energy harvesting system. This allowed us to create a model that, unlike the LCA, can produce a baseline green solution that can be used to interpret the results on the environmental impact of an IoT device. Furthermore, our proposal evaluates the environmental impact of the entire life-cycle of IoT devices. It is also worth noting that the proposed model can also be combined with those studies, such as [6,7], that evaluate the environmental impact of more complex platforms including network and cloud elements.

In particular, the studies in [6,7] propose models that assess the environmental impact generated by the use of network, fog and cloud devices. Fog [40] is a paradigm that refers to the distribution of storage and processing nodes to end-users. It allows overcoming the limits of a centralized solution (cloud). The exploitation of the nodes closest to the end-users is preferable with latency-sensitive applications, such as video streaming [41]. Ref. [6] proposes a flow-based model for estimating the energy consumption of shared network equipment and a time-based model for estimating the power consumption of customer premises equipment. Network equipment includes centralized datacenters, while fog devices are considered to be the user’s premises. In addition, the study evaluates the energy spent using an Ethernet, WiFi or 4G network in different application scenarios. Ref. [7] proposes a mathematical model to estimate the environmental impact in a hierarchical fog architecture. The authors describe the architecture in detail by evaluating the service latency, the power consumption and the CO${}_{2}$ emissions of real-time IoT applications by varying the number of connected devices and the ratio between the amount of data transmitted to the fog and the amount of data referred to the cloud.

Although many papers present guidelines, principles and definitions to achieve green IoT solutions, to the best of our knowledge, no study has yet formulated a model capable of considering not only the impact of the devices but also that of the maintenance and the energy harvesting system. Specifically, works [37,38,39] address the important problem of designing energy-efficient IoT solutions, but they do not focus on the impact of such solutions during the entire life-cycle; ref. [1] addresses the impact of information and communication technologies as a whole, but it does not have a specific focus on IoT; refs. [35,36] address the impact of smartphones; and refs. [6,7] address the impact of the network, fog and cloud infrastructures. Our work instead aims at making a step in the precise direction of an assessment of the impact of IoT solutions in their entire life-cycle, by introducing a model that estimates the impact on energy and disposal caused by IoT devices in an application. On the other hand, our work is just a step in this direction and it is subject to several limitations that we discuss in Section 8.2.

Table 1 summarizes the key differences between our approach and the closest related work.

## 4. System Model

The model proposed in this work aims at giving a complete accounting of the environmental impact of outdoor, energy harvesting IoT deployments considering the three phases of their life-cycle: manufacturing, utilization and disposal. To this purpose, we partition the environmental impact into energy impact and waste impact. The first concerns the manufacturing and utilization phases while the second concerns the disposal. We also take into account the impact of maintenance, because for reliable deployments with a large number of devices the maintenance may take a significant amount of resources and thus its environmental impact cannot be neglected. To make the deployment impact more comprehensible, the model provides the overall impact of an IoT solution as a ratio with respect to the impact of an equivalent baseline green solution. Note also that, for the sake of simplicity, our model assumes that all devices are sensors. However, it can be easily extended to other kinds of devices that include other specific components by adding parameters concerning the manufacturing, utilization and disposal of such components.

Note also that our model is modular: although it applies to a set of identical IoT devices, it can be used to analyze a complete IoT solution consisting of a heterogeneous set of devices by analyzing separately each subset of devices of the same type.

### 4.1. System Model Assumptions

In general, an IoT deployment comprises several devices that may differ on the type and number of communication, sensing and computing technologies and the kind of power supply solution.

For our purposes, we assume that a device consists of one or more boards, one or more sensors, one processor, a radio module and that it is equipped with an energy harvesting system (EHS). In outdoor IoT deployments, the use of EHS is a promising green strategy that simplifies the deployment and that contributes to reduce the size of the battery and extend the maintenance-free period. In particular, we consider the solar energy system because it is widely used being a mature and reliable technology capable of providing a great power density. We complete the energy system by considering a battery that is the energy storage of the system and that is characterized by a high percentage of non-recyclability and contains toxic materials. Note however that it is possible to extend our model for the use of other EHS. Such an extension would require just a value in megajoules (MJ) for the energy that the system can produce daily, an estimate in MJ that indicates the energy spent on its production and kilograms of waste produced when disposed. It is also possible to consider the devices powered by the main electricity grid by removing from the model all the factors relating to the solar panel and the battery.

To evaluate the impact of an IoT deployment, we consider these three components (device, solar panel and battery) together as a single system. Therefore, sense entities consist of *m* identical systems (Figure 2).

### 4.2. Life-Cycle Impact

To estimate the energy spent in the manufacturing process of a device, the model takes into account sensors, processor and radio module, considering them all as semiconductors. Hence, we calculate the energy spent on the production of an IoT device as the energy spent on its semiconductor elements. For the kilograms of waste produced by an IoT device, we consider the percentage of waste produced by its printed circuit boards (PCBs).

To the best of our knowledge, we are not aware of data provided by manufacturers of IoT devices concerning the energy spent on the production of their products or their percentage of recoverability. For this reason, our model estimates the manufacturing energy and waste impacts of the solar panels and batteries considering the studies of the respective sectors. The energy consumed in the utilization phase is equal to the difference between the daily load of the device ($E_{d_{i}}$) and the daily energy production of the EHS ($E(p_{i})$). Note that we consider that each device is powered by the energy harvesting system and the main electricity grid. This is a reasonable design choice that guarantees the functionality of the system for the whole day. In this way, we can model two cases:$E\left(p_{i}\right)<E\left(d_{i}\right)$: the solar panel is undersized and it is unable to produce enough energy to power the device for the whole day. In this case, $E\left(d_{i}\right)-E\left(p_{i}\right)$ will be the energy supplied by the main electricity grid.$E\left(p_{i}\right)=E\left(d_{i}\right)$: the solar panel can provide the exact amount of energy required by the device; therefore, in this case, the energy impact due to the use phase is 0.

There would be a third case, $E\left(p_{i}\right)>E\left(d_{i}\right)$, when the solar panel generates more energy than necessary. The extra power produced by the solar panel when the device is powered and the battery is fully charged is wasted; therefore, also in this case, the model considers the energy impact of the use phase equal to 0.

### 4.3. Maintenance

When designing an IoT deployment with replaceable batteries and high-precision sensors, we must take into account the maintenance to ensure a certain degree of reliability and a longer lifetime of the deployment. To this purpose, we assume an opportunistic time-based multi-component preventive maintenance where each component of the system is replaced before reaching the end of their life. We assign to each component a fixed estimate of the lifespan and maintenance is formulated as a 0–1 multi-objective linear programming problem in which the objective variables indicate whether a maintenance operation takes place at a specific time (1) or not (0). Note that we limit maintenance to a time-based strategy, but the model can be extended to include the most appropriate strategy for the application considered by changing the related equations to estimate the impacts of energy and waste. The energy impact of the maintenance refers not only to the production of the new components, but it also considers a fixed energy cost for the intervention of the technicians. In the same way, the waste impact takes into account the disposal of the components exhausted and a fixed waste impact for the intervention.

### 4.4. Baseline Green Solution

To facilitate the understanding of the environmental impact of an IoT deployment, the model calculates and compares the impact of the real deployment against the impact of a (theoretical) baseline green solution. Our model calculates this theoretical baseline green solution by keeping the device fixed and by resizing the solar panel and the battery. The dimensions of these two components are calculated to be the minimum necessary to guarantee the energy independence of the device in ideal conditions. Smaller size involves less impact in terms of energy and waste. Note however that in a real deployment, some degree of redundancy is needed. For example, a slightly larger solar panel is needed to ensure correct system behavior, even in the presence of changes in climate conditions.

Although it would be possible to adopt software optimizations or change the type and number of sensors, processors and communication technologies in order to devise a greener solution, this aspect is beyond the scope of this work and a matter of future investigations. On the other hand, resizing the components of the energy harvesting system is a practice that can be applied to all deployments that exploit an energy harvesting solution. Note that, if different devices are available, the greenest one can be estimated by applying the model once for each of them.

## 5. Estimating the Impact of IoT Deployments

The model consists of two main parts: an environmental impact model and a baseline green model. The first one defines the impact of the energy and waste of both devices and the energy harvesting components of an IoT deployment. The second one defines a baseline green solution, estimates its impact and provides the ratio between the two solutions. Such a ratio provides an estimate of the efficiency of a deployment against its baseline, considering both the consumed energy and produced waste.

### 5.1. Environmental Impact Model

The environmental impact model defines the impact, in terms of consumed energy and produced waste, of an IoT deployment during its entire lifetime.

Equation (Equation 1) estimates the energy $E_{S}$ consumed by an IoT deployment *S* by considering the energy spent for the production of all its *m* IoT devices ($P_{d_{i}}$), solar panels ($P_{p_{i}}$) and batteries ($P_{b_{i}}$), the energy $U_{d_{i}}$ spent for the utilization of the devices during their operation life and the energy *PM*${}_{sp}$ spent for the production and maintenance of the spare components used during maintenance:(1)$$\begin{array}{c} E_{S}=\sum_{i=1}^{m}\left(P_{d_{i}}+P_{p_{i}}+P_{b_{i}}+U_{d_{i}}+{PM}_{sp}\right) \end{array}$$

Equation (Equation 2) estimates the waste $W_{S}$ caused by an IoT deployment *S* by considering the waste produced by the disposal of the *m* deployed IoT devices ($D_{d_{i}}$), solar panels ($D_{p_{i}}$) and batteries ($D_{b_{i}}$), as well as the waste $M_{sp}$ caused by the spare components used during maintenance:(2)$$W_{S}=\sum_{i=1}^{m}\left(D_{d_{i}}+D_{p_{i}}+D_{b_{i}}+M_{sp}\right)$$

Equations (Equation 1) and (Equation 2) aim at defining a general model for estimating the environmental impact of a deployment, taking into account all the elements of an IoT deployment.

We discuss next how each element of Equations (Equation 1) and (Equation 2) can be estimated. We organize the discussion along the phases of a system life-cycle: production, utilization, disposal and maintenance.

#### 5.1.1. Production

The energy spent for the manufacturing of devices ($P_{d_{i}}$), solar panels ($P_{p_{i}}$) and batteries ($P_{b_{i}}$) is estimated with Equations (Equation 3)–(Equation 5): (3)$$\begin{array}{c} P_{d_{i}}=e_{d}\times \sum_{j=1}^{n_{i}}size\left(el_{j}\right) \end{array}$$(4)$$\begin{array}{c} P_{p_{i}}=e_{p}\times size\left(p_{i}\right) \end{array}$$(5)$$\begin{array}{c} P_{b_{i}}=e_{b}\times weight\left(b_{i}\right) \end{array}$$
where $e_{d}$, $e_{p}$ and $e_{b}$ denote the average energy needed to produce 1 cm${}^{2}$ of semiconductor elements, 1 m${}^{2}$ of solar panels and 1 kg of batteries, and where $size(el_{j})$, $size(p_{i})$ and $weight(b_{i})$ denote the number of semiconductor elements, the solar panel surface and the battery weight for the *i*-th device.

To estimate the energy spent for manufacturing a component, the specific production method should be considered (see Section 2 for an analysis of the literature on production methods). For each component, we consider an average energy consumption according to the characteristics and technology of that component. For example, the energy needed to produce a solar panel depends both on its surface and on the employed technology (mono-Si or poly-Si). Due to the heterogeneity of the components involved and the different production processes, we employ the following average values extracted from [16,17,20]:
$e_{d}\;\left({\mathrm{MJ}/\mathrm{cm}}^{2}\right)$$e_{p}\;\left({\mathrm{MJ}/m}^{2}\right)$$e_{b}\;\left(\mathrm{MJ}/\mathrm{kg}\right)$Semicond.mono-Sipoly-SiLithiumNiMh5.5445476.14676.1164.8204.143

#### 5.1.2. Utilization

The energy $U_{d_{i}}$ spent for the utilization of the devices during their operation life is estimated with Equation (Equation 6):(6)$$\begin{array}{c} U_{d_{i}}=\left\{\begin{array}{c} {life}_{S}\times \left(E\left(d_{i}\right)-E\left(p_{i}\right)\right)\;\;\;\mathrm{if}\;E\left(d_{i}\right)>E\left(p_{i}\right)\\ 0\;\;\;\;\;\;\;\;\;\;\;\;\;\;\;\;\;\;\;\;\;\;\;\;\;\;\;\;\;\;\;\;\;\;\;\;\;\;\mathrm{otherwise} \end{array}\right. \end{array}$$
where *life*${}_{S}$ denotes the life duration of deployment *S*, $E(d_{i})$ the energy consumed by device $d_{i}$ for its use and $E(p_{i})$ the energy produced by its solar panel $p_{i}$.

If the energy $E(d_{i})$ consumed by device $d_{i}$ is greater than the energy $E(p_{i})$ produced by its solar panel $p_{i}$, we assume that the missing energy will be supplied by the main electricity grid. If instead $E\left(d_{i}\right)\leq E\left(p_{i}\right)$, then the device is self-powered as the energy harvesting system produces enough energy to power the device.

Many factors influence the energy required by a device, such as the message transfer rate, the type of communication network and the energy consumption of the individual elements of the device. In general, these devices cyclically perform the same operations (sense–process–communicate) and remain in sleep mode most of the time. They are characterized by a short duty cycle to improve battery lifetime.

We model the energy $E(d_{i})$ required by a device $d_{i}$ for its utilization with Equation (Equation 7):(7)$$\begin{array}{c} E\left(d_{i}\right)=\sum_{j=1}^{n_{i}}\left(dc_{j}\times \eta_{A}\left(el_{j}\right)\right)+\left(\left(1-dc_{j}\right)\times \eta_{I}\left(el_{j}\right)\right) \end{array}$$
where $dc_{j}$ denotes the length of the duty cycle of j-th element of *i*-th device, and $\eta_{A}\left(el_{j}\right)$ and $\eta_{I}\left(el_{j}\right)$ denote the power consumption of j-th element when active and when inactive (sleep mode), respectively.

We model the energy $E(p_{i})$ produced by a solar panel $p_{i}$ with Equation (Equation 8):(8)$$\begin{array}{c} E\left(p_{i}\right)=size\left(p_{i}\right)\times \epsilon \left(p_{i}\right)\times \epsilon \left(b_{i}\right)\times \rho \end{array}$$
where $\epsilon (p_{i})$ and $\epsilon (b_{i})$ denote the efficiency of solar panel $p_{i}$ and of battery $b_{i}$, respectively, and ρ the daily irradiation (MJ/m${}^{2}$), which can be estimated with tools such as PVGIS [42].

#### 5.1.3. Disposal

We consider the impact of waste when a component reaches the end of its life. The non-recoverable percentage of a component depends on its technology and can be estimated from the weight of that component. We calculate this impact in kg, but it is worth noting that such weight includes hazardous and toxic materials. While a single component does not have a relevant impact on the environment, 22 billion devices do [43].

To estimate the amount of waste produced by the components of an IoT solution, we consider the percentage of components that cannot be recycled or reused by the recovery process. While Section 2 discussed the recovery processes of devices, solar panels and batteries, Equations (Equation 9)–(Equation 11) provide an estimate for them (in kg): (9)$$\begin{array}{c} D_{d_{i}}=w_{d}\times \sum_{j=1}^{n_{i}}weight\left(pcb_{j}\right) \end{array}$$(10)$$\begin{array}{c} D_{p_{i}}=w_{p}\times weight\left(p_{i}\right) \end{array}$$(11)$$\begin{array}{c} D_{b_{i}}=w_{b}\times weight\left(b_{i}\right) \end{array}$$
where $w_{d}$, $w_{p}$ and $w_{b}$ denote the average waste percentage of devices, solar panels and batteries, respectively, and where $pcb_{j}$ denotes the *j*-th printed circuit board of the *i*-th device.

As waste percentages vary depending on the employed technology, we extracted average waste percentage values from [13,14,18,22,23]:
$w_{d}$$w_{p}$$w_{b}$Semicond.mono-Sipoly-SiLithiumNiMh38%16%16%55%67%

It is easy to see that batteries are the less recyclable components—and the less reliable elements—whose duration depends on many factors, such as temperature, type of load and technology.

#### 5.1.4. Maintenance

While most existing studies on IoT deployments do not consider maintenance activities, they are necessary to ensure the reliability of any deployment, and they do contribute to the consumed energy and generated waste.

We model the energy *PM*${}_{sp}$ spent for the production and maintenance of the spare components used for maintenance interventions, and the waste $M_{sp}$ induced by those components, as a multi-objective linear programming problem (MOLP), defined by two objective functions: (12)$$\begin{array}{c} {minimize\;PM}_{sp}=\sum_{t=1}^{T}\left(x\left(i,t\right)\times p\left(i\right)+y\left(t\right)\times {fe}_{t}\right) \end{array}$$(13)$$\begin{array}{c} {minimize\;M}_{sp}=\sum_{t=1}^{T}\left(x\left(i,t\right)\times w\left(i\right)+y\left(t\right)\times {fw}_{t}\right) \end{array}$$
where



$x\left(i,t\right)=\left\{\begin{array}{c} 1\;\;\;\mathrm{if}\;\mathrm{component}\;i\;\mathrm{is}\;\mathrm{replaced}\;\mathrm{at}\;\mathrm{time}\;t;\\ 0\;\;\;\mathrm{otherwise}; \end{array}\right.$

p(i) denotes the energy cost for producing a new component *i* (see Equations (Equation 3)–(Equation 5));

$y\left(i\right)=\left\{\begin{array}{c} 1\;\;\;\mathrm{if}\;\mathrm{maintenance}\;\mathrm{occurs}\;\mathrm{at}\;\mathrm{time}\;t;\\ 0\;\;\;\mathrm{otherwise}; \end{array}\right.$

w(i) denotes the waste cost for the disposal of component *i* (see Equations (Equation 9)–(Equation 11));${fe}_{t}$ and ${fw}_{t}$ denote a fixed energy and waste cost of a maintenance intervention at time *t*.

We assume that maintenance interventions are scheduled over a set of *T* time intervals across the life duration of the deployment, and that they may not require replacing components, namely:(14)$$\begin{array}{c} \forall i,t:x(i,t)\in \{0,1\}\;\wedge \;y(t)\in \{0,1\}\;\wedge \;x(i,t)\leq y(t) \end{array}$$

We also assume that each component will be replaced at least once during the deployment lifetime, namely:(15)$$\begin{array}{c} \sum_{t=1}^{T}x\left(i,t\right)\geq 1 \end{array}$$

The output of this model is the time-schedule of the maintenance which allows obtaining the best trade-off between the values of the two objective functions. Once all the parameters are defined, it is possible to solve the multi-objective linear programming (MOLP) problem, obtaining the energy and waste impact of the maintenance.

The output of this model is the best trade-off between the minimum energy spent and the minimum waste produced for the maintenance phase. The IoT impact calculator web service presented in Section 7 is an easy and fast instrument that helps to define each parameter and automatically solve the MOLP problem.

For what concerns the time complexity of the model, we observe that it is dominated by the MOLP problem of Equations (Equation 12) and (Equation 13). Although this problem is known to be NP-complete, there are several solvers available that can find a solution for problems of moderate size, even on standard desktop PCs. In the evaluation of our case study in Section 6, we used the Gurobi Python library [44].

### 5.2. Baseline Green Solution

To estimate the efficiency of an IoT deployment, we define a baseline green solution, estimate its impact (consumed energy and produced waste) and consider the ratio between the deployment and the baseline solution.

The baseline model resizes the energy harvesting system to reduce the energy required for its production and to decrease the waste generated by its end of life. Moreover, the baseline sizes solar panels and batteries on the basis of the daily energy required by the devices, so as to get zero environmental impact during the utilization phase.

The size $size(p_{i})$ of a solar panel $p_{i}$ capable of providing the energy $E(d_{i})$ required daily by a device $d_{i}$ is defined as:(16)$$\begin{array}{c} size\left(p_{i}\right)=\frac{E(d_{i})}{\epsilon \left(p_{i}\right)\times \epsilon \left(b_{i}\right)\times \epsilon \left(r_{i}\right)\times wo\left(p_{i}\right)\times \rho } \end{array}$$
where $\epsilon (p_{i})$, $\epsilon (b_{i})$, $\epsilon (r_{i})$ denote, respectively, the efficiency of solar panel $p_{i}$, of battery $b_{i}$ and of the output regulator $r_{i}$; $wo(p_{i})$ the wear-out of $p_{i}$; and ρ the daily irradiation.

It is worth noting that the wear-out of a solar panel must be taken into account, as it makes the panel less efficient over the years. For instance, after 20 years, a solar panel with constant annual degradation rate of 1% will produce only 80% of its initial capacity [19], and it may hence become insufficient to power a device.

Secondly, a battery must have a capacity at least equal to that required by the device in the dark hours of a day. The minimum capacity of a battery is hence defined as:(17)$$\begin{array}{c} capacity\left(b_{i}\right)=\frac{E(d_{i})\times \delta }{\epsilon \left(b_{i}\right)\times \epsilon \left(r_{i}\right)} \end{array}$$
where δ indicates the percentage of hours without sun, when only the battery will power the device.

As we have seen in the previous section, solar panel and battery are the essential components for an energy harvesting system. Other elements can be introduced and their energy consumption can be considered in the design by adding their efficiency in the denominator of Equation (Equation 16).

As discussed in the previous section, we evaluate the waste impact of the components based on their weights. To estimate the weight of solar panel and batteries, we use their power density: (18)$$\begin{array}{c} weight\left(p_{i}\right)=capacity\left(p_{i}\right)\times density\left(p_{i}\right) \end{array}$$(19)$$\begin{array}{c} weight\left(b_{i}\right)=capacity\left(b_{i}\right)\times density\left(b_{i}\right) \end{array}$$
where $capacity(p_{i})$ denotes the nominal capacity of the solar panel in ideal conditions [18], $density(p_{i})$ the density of the solar panel (expressed in kg/Wp) and $density(b_{i})$ the density of the battery (expressed in kg/MJ).

The nominal capacity of the solar panel can be calculated as $\epsilon \left(p_{i}\right)\times size\left(p_{i}\right)$. We considered crystalline solar panels with a density of 0.102 kg/Wp [18]. The density of the battery can be computed from the battery employed in the deployment as the ratio of its weight and capacity ($capacity\left(b_{i}\right)/density\left(b_{i}\right)$).

We can now apply Equations (Equation 1) and (Equation 2) to obtain the environmental impact of the baseline solution. The efficiency of a deployment *S* can now be assessed against the baseline green solution *G*, namely:(20)$$\begin{array}{c} \nu_{energy}=\frac{E_{S}}{E_{G}}\;\;\;\;\;\;\;\;\;\;\;\;\;\;\;\nu_{waste}=\frac{W_{S}}{W_{G}} \end{array}$$

The closer the ratios $\nu_{energy}$ and $\nu_{waste}$ to 1, the more efficient the assessed deployment.

## 6. Case Study

We now illustrate the effectiveness of our model by analyzing its application to a real IoT deployment. To this purpose, we selected the IoT deployment implemented in the RESCATAME European project, which includes a significant number of solar power, energy harvesting IoT devices deployed in the Spanish city of Salamanca. This analysis is organized in four parts. In the first part (Section 6.1), after briefly revising the main aspect of the project, we assess the impact of its devices, energy harvesting subsystem and maintenance, and we conclude this section by presenting the aggregate results of this analysis. In the second part (Section 6.2), we present a baseline green solution which is functionally equivalent to that used for RESCATAME, but with a smaller impact, and in the third part (Section 6.3), we compare the energy and waste impact of the RESCTAME and of its baseline green solution. The last part (Section 6.4), still referring to the RESCATAME deployment, discusses the effect of different geographical locations at different latitudes in terms of the energy and waste impact.

### 6.1. Rescatame Project Green Analysis

The RESCATAME project was run in Salamanca in 2010 and was the first full-scale implementation of an air quality sensor network for urban traffic management [9]. The implementation involves the use of 35 sensor devices deployed on two roads with a high traffic density: Calle Álvaro Gil with 10 devices and Avenida de los Cipreses with 25 devices. The devices acquire information related to the levels of carbon monoxide (CO), nitrogen dioxide (NO_2_), ozone (O_3_), particulates (PM), noise, humidity and temperature in the air. The acquired data are sent via Zigbee to a router that re-routes this information to a database via GPRS or Ethernet. The proposed solution provides accurate measurements of real-time air quality and calculates forecasts on pollution levels with high reliability. The forecasts allow the traffic department to perform some actions to guarantee a certain quality of the air. All devices are equipped with an energy harvesting solution composed of a solar panel and a rechargeable lithium battery. Both the routers and devices were Libelium products [45].

We first assess the impact of each device of the RESCATAME system, then we proceed by analyzing the impact of the maintenance and then of the whole solution in four application lifespan scenarios.

#### 6.1.1. Devices

To estimate the impact of a single device in use, we obtain the data about its components from the datasheets and the studies on this project [46,47]. The project adopts Waspomote boards equipped with dedicated boards to connect the seven sensors for the acquisition of the air quality data. Table 2 summarizes the specifications of all the components of a device. In particular, to estimate the energy impact, we need the area and the power in the active and sleep mode of each semiconductor element of the device. To estimate the impact of the waste, only the weight of the boards is needed. The two boards have the same weight, which in total accounts for 0.04 kg.

By applying our model, we obtain the environmental impact of the device in its life-cycle:Manufacturing: knowing the area of each element and the average value for the manufacturing of 1 cm${}^{2}$ of semiconductor, we can apply Equation (Equation 3). An energy impact equal to 147.72 MJ per device is obtained.Utilization: in this project, the devices are active once every 10 min. To estimate the daily energy demand to run, we compute the average value, considering a duty cycle of 5% for all the device elements. Table 2 shows the current consumption in active and sleep mode and we set a voltage of 3.3 V. Exploiting Equation (Equation 7), we obtain the average energy required of $1.22\times 10^{-3}\;\frac{\mathrm{MJ}}{\mathrm{day}}$.Disposal: the percentage of non-recyclability of a board was estimated at 38%, and through Equation (Equation 9), an impact of 0.015 kg is obtained for each device.

#### 6.1.2. Energy Harvesting System

The RESCATAME project employs an energy harvesting system consisting of a monocrystalline photovoltaic (PV) panel and a lithium battery. In particular, the PV panel is characterized by an efficiency of 17%, a weight of 0.54 kg and its dimensions are 0.23 × 0.16 m. The battery has a capacity of 6600 mAh and weighs 0.16 kg. This system allows the device to be energetically autonomous by satisfying the load required by Waspmote and the sensors board. Because the method for estimating the impact of both of these components is the same, we show their analysis together.

Manufacturing: considering Equation (Equation 4) for the solar panel, we need the average energy consumed to produce 1 m${}^{2}$ of a PV panel and the area of the panel employed. Similarly, Equation (Equation 5) allows estimating the manufacturing energy of a battery using the average value to produce 1 kg of the storage system and the weight of the battery used. Solving both equations, we obtain a manufacturing impact of 205.03 MJ for the solar panel and 25.54 MJ for the lithium battery.Utilization: using Equation (Equation 8), we estimate the daily production of the solar panel in Salamanca to be 0.03 MJ. According to the PVGIS tool, the solar irradiation in the darkest month in Salamanca is January, and its daily energy irradiation is $4.06\;\frac{\mathrm{MJ}}{m^{2}}$. The energy produced by the solar panel is much higher than the amount required by the device. However, we must consider all the inefficiencies introduced by the system components. We assume 80% [19] for the efficiency of the solar panel at the 20th year, 90% [20] for the efficiency of the lithium battery and 90% [57] for the efficiency of the output regulator. Considering these three efficiencies, from the initial 0.03 MJ, the device will have 0.02 MJ which is still more than sufficient.Disposal: assuming as the non-recyclability factor 16.02 and 55.2%, respectively, for the monocrystalline solar panel and the lithium battery, the components of this solution will generate a total of 0.11 kg of waste. It must be considered that the battery is both the most toxic and the shortest-lived component. This implies that it will be replaced more frequently than the others, generating 0.09 kg of waste each time.

Table 3 reports the impacts in the three life-cycle phases of the IoT system considered. It can be seen that the energy harvesting system is oversized compared to the device load. The solar panel produces 21.23 times the daily energy required by the device, while the battery can fully power the device for about two months. To ensure a high degree of reliability, it is reasonable to increase the size of the energy harvesting system. However, oversizing involves a greater environmental impact; therefore, it is important to size all the components, also paying attention to the impacts of the energy and waste.

#### 6.1.3. Maintenance

To facilitate the calculations and reasoning, we consider the two roads (Calle Álvaro Gil and Avenida de Los Cipreses) as two independent deployments. A detailed analysis is proposed for Calle Álvaro Gil (10 devices). For this project, we assume that an intervention has only an energy impact and this is modeled as the energy spent by the maintenance technician to reach the devices by car. To estimate this impact, we calculate an average route considering the area of Salamanca (38.6 km${}^{2}$) and taking its radius (3.51 km). Assuming a fuel consumption of 6 l/100 km and a conversion factor of 8.9 kWh [58] for one liter of gasoline, we obtain 6.74 MJ spent for each intervention. The expected service life of each system component must be defined. For the solar panel, we assume a life of 20 years [19,59] which is a value used in many studies and is generally the warranty period. For a lithium battery, predicting its lifetime is more complicated. The battery is discharged daily by a non-constant load and is affected by the external temperature. According to the study [20], a lithium battery can last about 15 years if we consider a cycle per day. We reduce this time to 9 years taking into account the behavior of a lithium battery at low temperatures [27]. The evolution of gas sensors (CO, NO_2_ and O_3_) has made it possible to have smaller, cheaper and longer-lasting sensors; we set a duration of 10 years for these three sensors [60,61]. The life of the remaining sensors and the radio module is considered equal to the microcontroller unit (MCU). We refer to these components as the Motherboard. These elements are active only 5% of the time, but we did not find precise studies that estimate the life of an MCU, so we have used the data published by some producers. From the tests performed by Microchip and Texas Instruments [62,63], we can assume that these devices have a long life, longer than that of the application. Having the impact of one maintenance intervention, the duration of each component and their impacts of energy and waste, we formulate the problem of the optimization of maintenance as a linear programming problem. Through the multi-objective linear programming problem, and by means of a python solver based on the Gurobi Python library, we find the optimal scheduling strategies to minimize the energy and waste impacts considering 15, 20, 25 and 30 years as the application lifespan scenarios.

The maintenance problem follows the figures reported in Equations (Equation 12) and (Equation 13), where m=10 systems, *T* = 15, 20, 25 and 30 time intervals. The maintenance energy cost for one system is ${fe}_{t}=0.67$. Table 4 reports the impacts of the energy and waste due to maintenance in the four lifespan scenarios.

#### 6.1.4. Results

From the study, we obtain that, for the system in Calle Álvaro Gil, the energy spent on the production and maintenance per device ranges from 415.25 MJ with 15 years of lifespan to 684.8 MJ in the case of 30 years. If decommissioned after 15 years, a system will generate 0.27 kg of waste, and after 30 years, 0.53 kg. Now, applying the model for Avenida de Los Cipreses road, we are able to estimate the impacts of the total deployment (35 devices) by adding together the results obtained by the two roads.

The energy required by the entire deployment ($E_{S}$) ranges from 14,523 MJ in the case of a 15-year lifespan to 23,917 MJ in the 30-years case (Figure 3) (note that in this case, the maintenance does not scale linearly with the number of devices). The total amount of waste produced by the systems ($W_{S}$) in the different scenarios ranges from 9.55 to 18.58 kg in the cases of 15 and 30 years (Figure 4). As the lifespan increases, the impact of the maintenance also increases due to the greater number of replaced components. Maintenance becomes the main contributor to the energy (42%) when the solar panel needs to be replaced and is the main contributor to waste (65%), having to frequently replace the battery. This underlines the importance of correctly sizing the energy harvesting system as large components imply greater impacts. It should be noted that without a maintenance strategy, in the case of a duration of 30 years, we should re-deploy all systems every 15 years while through this maintenance strategy it is possible to save 18% of the energy and 3% of the waste.

### 6.2. A Baseline Green Solution for RESCATAME

We have seen that the solar panel and the battery of the RESCATAME implementation are oversized, in particular the solar panel is able to produce about 21.23 times the amount of energy required by the device every day and the battery could power the device for about two months. In a more ecological but ideal design choice, the components are able to provide the exact amount of energy required by the device considering all the inefficiencies introduced by the various system components (these inefficiencies are shown in Table 5). Our model estimates the impact of this ideal design as the baseline green solution, by means of Equations (Equation 16) and (Equation 17).

By keeping the device load and impacts fixed, we resize the solar panel and the battery to provide the daily energy required by the device. From the model, we obtain a solar panel 13.75 times smaller than that supplied by Libelium and a lithium battery with a capacity of $0.94\times 10^{-3}$ MJ. Having a new energy harvesting system, we can apply the figures computed in the previous section to obtain the impacts of the energy, maintenance and waste.

The total energy ($E_{G}$) spent for the production and maintenance of the baseline green solution considering all the 35 systems varies from 6102.76 to 7075.65 MJ while the waste produced ($W_{G}$) in the 15 and 30 years cases is 0.87 and 1.21 kg. The main contributor in this solution in terms of the energy spent and waste produced is the device, which is the component that we have assumed to be fixed.

### 6.3. Comparison of RESCATAME with the Baseline Green Solution

The analysis shows that the energy impact of the implementation of the RESCATAME project ranges between 14,523 MJ in the case of the 15-years lifespan and 23,917 MJ in the 30-years scenario. The kilograms of waste varies in these two cases from 9.55 to 18.58 kg, respectively, for 15 and 30 years. In order to assess how much the RESCATAME implementation is green, we compare these figures against those of the baseline green solution (Figure 5 and Figure 6).

In Table 6, $\nu_{energy}$ indicates the ratio of the energy impact between the two solutions and goes from 2.4 in the case of the 15-years lifespan to 3.4 in the 30-years case. $\nu_{waste}$ (Table 7) is the ratio of the waste impact and ranges between 11 and 15.4 in the 15- and 30-years scenarios, respectively. So, the difference between the two solutions increases as the life of the application increases. This is due to the greater number of components replaced and the different size of the energy harvesting system components. This is the consequence of the oversizing of the components in the design of the RESCATAME system.

We now analyze in deeper detail the impact in the case in which the system life is 30 years. The baseline consumes 70.41% less energy than the real implementation and produces 93.51% less waste. The solar panel is the system component that requires more energy for the manufacturing and produces more waste once it reaches the end of life. As already discussed, this is due to its dimensions that are oversized. In the baseline, the solar panel weighs much less (Figure 7 and Figure 8).

Looking at the maintenance impact in detail, we found that the solar panel in the RESCATAME implementation represents 67% of the impact on the maintenance energy (Figure 9). In maintaining the baseline green solution, the costs are more balanced indeed; the maintenance of the solar panel and battery weighs 40%. The difference between the two solutions is more evident considering the waste. Looking at the configuration of the waste produced by the maintenance (Figure 10), we can see how the weight of the battery is significantly reduced in the baseline, which is an important factor because this component is the least recyclable and the most toxic.

### 6.4. Green Deployments and Latitude

The model proposed in Section 5 allows to properly size the energy harvesting solution to meet the daily energy required by the device. Although this energy is constant even if we deploy the RESCATAME solution in another place, the parameters computed for the solar panel and the battery may no longer be sufficient. It is interesting to see how the impact of the baseline green solution changes if implemented in the cities at a latitude above Salamanca. The cities considered are London, Edinburgh, Glasgow and Oslo. These cities have different levels of solar radiation and hours of light compared to those of Salamanca and these parameters are necessary to correctly size the energy harvesting system components. Using the tool [42], we can determine the solar radiation on the darkest day, and through the data reported by [65], we determine the hours of light at each day of the year in each city.

To analyze how the impact of the green solution varies according to the latitude at which it is implemented, we apply the model by considering the deployment described in the previous sections, along with the technologies and inefficiencies considered so far (see Table 5), except for ρ (daily irradiation) and δ (percentage of hours without sun) which vary according to the city considered. For each city, Table 8 shows the dimension of the solar panel and the battery needed to produce $1.22\times 10^{-3}\;\frac{\mathrm{MJ}}{\mathrm{day}}$ which is the daily energy required by the device.

As expected, the dimension of the components increases with the decreasing of the irradiance and the number of light hours. Moving from the south to the north, the size of the solar panel roughly doubles from one city to the other. The increase in the battery capacity is less evident. The battery required in Oslo is 1.27 times larger than the one used in Salamanca, and this is due to the difference in the number of hours without light which goes from 62.5 to 79.17% for Salamanca and Oslo, respectively. Considering these new energy harvesting systems, we can analyze how the energy and waste impact of the entire deployment vary. Larger components have a greater impact, this is inevitable, but it is interesting to analyze the weight of these larger components.

Figure 11 shows the energy impact of the solution deployed in the cities considered while Table 9 reports the ratio of their impacts. Oslo has the greatest impact due to the larger components, while Salamanca consumes less energy by having more hours of light and higher irradiance. The energy impact in the case of Oslo is 2.45 greater than the Salamanca impact. The same behavior can be seen in the waste, Figure 12 and Table 10. The green solution produces in Oslo 5 times the waste that the same solution produces in Salamanca. It is interesting to note that the impacts derived from the Oslo solar panel are 10.82 times the impacts of Salamanca. Instead, the impacts of the battery are not so far from those of the Spanish city. Indeed, the energy impact and the waste impact of the battery in Salamanca are 10.71 MJ and 0.04 kg, respectively, while in Oslo they are 13.56 MJ and 0.05 kg.

Now, we can compare the environmental impact of the implementation of RESCATAME with the impact of the baseline in the different cities. We can notice that although moving toward the cities of the north the gap between the impacts decreases, the energy impact of the baseline in Oslo is 27.53% lower than the real implementation, saving 6583.775 MJ. The growth trend of the waste is slower than the energy consumption and this is because the battery is very oversized in the project. The waste produced by the baseline in Oslo is 67.53% less than that produced by RESCATAME, allowing the waste impact of Oslo to remain far below the impact of RESCATAME.

## 7. IoT Impact Calculator

The proposed model provides a complete view of the environmental impact of an IoT deployment considering the impact in each phase of the life-cycle of the system components. The IoT impact calculator web service is a practical and useful instrument to make using the model simple and to obtain clear results. The implementation of the web service is available on GitHub. The frontend is available at https://github.com/edoBaldini/vue-g-iot-calc (accessed on 3 September 2021), The backend is available at https://github.com/edoBaldini/Rest-G-IoT-Calc-API (accessed on 3 September 2021).

Figure 13 sketches the architecture of the web service. The backend developed through the Flask [66] framework reflects the model presented in the previous section. To solve the multi-objective linear programming problem, we used the Gurobi Python library [44].

To increase the user experience, we implemented the user interface (UI) as a responsive single-page web app using Vue.js [67] and Bootstrap [68], making the service user-friendly.

The UI guides the user through the evaluation process. It is based on a multi-step form in which the impact assessment of a single component is performed at each step. For those parameters that are not easy for the user to retrieve, the IoT impact calculator provides default values obtained from the literature. The last step of the form displays the environmental impact of the user solution and the impact of the baseline green solution. The impact is expressed in terms of the energy for the manufacturing and use phase and in kilograms for the disposal phase. The impact is shown through graphs; in this way, the distance between the user’s solution and the baseline proposal is evident and easy to understand. At the bottom part of the page (Figure 14), the ratios between the user and baseline impact are reported.

The implementation of the IoT impact calculator makes the model easily usable and accessible by any user.

## 8. Discussion and Conclusions

As concluding remarks, we summarize the contribution of this work (Section 8.1), we thoroughly discuss the limitations of our approach (Section 8.2) and, finally, we present our ongoing and future work on this research area (Section 8.3).

### 8.1. Contributions

The model proposed in this work aims at assessing the environmental impact of an IoT deployment considering the use of a solar energy harvesting system. The available literature generally focuses on the energy spent on production or on the waste generated by a single component (device, solar panel and battery). Other studies report the deployment of IoT solutions with the energy harvesting systems, without considering the impact due to the production and disposal of the components used nor the influence of maintenance.

Our work instead contributes to the definition of a model aimed at the following:Assessing the impacts of each component in a system (device, solar panel and battery) by considering the production phase, the use phase and the end of life, thus covering its entire life-cycle. From the manufacturing and use phases, the model returns the estimate of the energy consumed, and from the end of life, it provides the waste impact. The model evaluates the environmental impact of the maintenance considering a time-based preventive maintenance strategy. This is formulated as a multi-objective mixed-integer linear programming problem. The model returns a value in megajoules (MJ) for the energy impact of the entire deployment and in kg for the waste impact.Computing a baseline green solution by reconsidering the size of the solar panel and the battery. The baseline is defined as a functionally equivalent device operating in ideal conditions, and thus with a reduced panel and battery, but still guaranteeing the energetic autonomy of the system. The model evaluates the impacts of the baseline, and it shows the ratio between the energy impact and the waste impact of an IoT deployment against its baseline. Through the ratio, it is possible to interpret the impact of an IoT deployment by comparing it against its baseline.

To make the model easy to use by others, we developed the IoT impact calculator web service, which is an open-source prototype to evaluate a solution and to analyze, through diagrams, the differences between the considered solution and its green baseline.

To illustrate the use of the model, we presented an extensive study that applies to a large outdoor, energy harvesting deployment developed within the European project RESCATAME [9]. Please note that this study aims only at showing the use of the model, while extensive experimentation is currently a matter of future work. In particular, this study evaluates the environmental impact of this deployment in four different lifespan scenarios (15/20/25/30 years). In the last scenario, the model estimates that the RESCATAME deployment consumes more than 3 times the energy required by its baseline green solution and produces waste for a volume 15 times greater. To show how the environmental impact of the baseline relates to the latitude of the place of installation, we applied the model considering different possible deployments in cities at different latitudes (namely the cities of Salamanca, London, Edinburgh, Glasgow and Oslo). We thus analyzed how, moving toward the cities of northern Europe, the gap between the impacts of the RESCATAME implementation and its baseline decreases. The ratios between the impacts of Oslo and the RESCATAME were 1.4 for the energy impact and 3.1 for the waste impact, demonstrating that even in cities in northern countries (or southern countries in the southern hemisphere) it is possible to significantly reduce the environmental impact by correctly sizing the components of an IoT system.

### 8.2. Limitations

Because the IoT is an emerging technology, it is difficult to find precise and up-to-date data relating to the energy consumed for their production and the impact of their disposal. The literature is sparse and manufacturers do not provide, to the best of our knowledge, a life-cycle assessment for a single device. This led us to estimate the energy spent to produce a device by considering only its semiconductor elements and printed circuit boards to evaluate the impact of the waste.

Secondly, this work considers only solar energy harvesting. On the other hand, this is also one of the most widely used technologies and the impact of the production and disposal of the photovoltaic panel and battery is well documented. However, concerning this last aspect, a further limitation of the present work is in the accuracy of the estimated impact. Indeed, most solar panel studies consider panels much larger than those used in IoT deployments. Although their material composition is the same, it is difficult to find specific information for small panels. The same consideration applies also to batteries, as very few papers study small batteries with periodic loads, which is actually the case of many IoT deployments. A further limitation of the present work is that also other harvesting systems are available for IoT devices. As shown in [69], other energy sources may be thermal, radio-frequency or piezoelectric. Hence, further studies are needed to assess the environmental impact of all these energy harvesting systems.

Furthermore, our model addresses basically IoT devices with energy harvesting and sensing subsystems. Although an IoT deployment may also include other devices (such as actuators or networking/storage-oriented devices), we limit to these (relatively) simple devices. Note however that network and storage devices have been the subject of different studies [1,6,70], and our model can be easily extended to include these other devices by adding parameters concerning the environmental impact in their manufacturing, use and disposal phases.

Finally, we disregard aspects such as recycling, the impact of environmental and operative conditions on the lifetime of the devices and components and the costs of deployment that may depend on the location or the environment of the deployment itself.

In conclusion, the limitations of the present work mainly concern the accuracy of some data. However, due to its flexibility and modularity, the proposed model can be easily improved with more accurate parameters and more precise equations, if available, and also with other equations to evaluate the impact of new components different than those considered.

### 8.3. Current and Future Work

The main objective for the evolution of the proposed model is to overcome the limitations discussed in the previous section. In addition to improving the accuracy of the data, which is the main limitation, future work may aim at introducing a green label. IoT deployments are used in many different applications, including Smart Homes, e-Health and smart factories [71], but they are also used to implement smart wearable devices, so they are also consumer products. In all these applications, values in MJ and in kg can be difficult to interpret, and for this reason, we have provided the ratios between the evaluated solution and its baseline. A natural long-term evolution of this concept is the definition of a green label. An environmental efficiency label would allow the customers to choose the IoT solution that better exploits the energy, allowing them to save money and contributing less to the global impact. We can take as an example the way in which the energy efficiency class ($\eta_{TM}$) in the labeling of the light bulbs [72] is computed:(21)$$\begin{array}{cc} \eta_{TM} & =\left(\frac{\phi_{use}}{P_{on}}\right)\times F_{TM} \end{array}$$
where $\phi_{use}$ is the nominal useful luminous flux, $P_{on}$ is the nominal power consumption in active mode and $F_{TM}$ is a factor that changes according to the type of the light source, whether directional or not and whether the light is connected to the mains or not. The obtained value is compared with fixed parameters to frame the value on the scale. If the value obtained is equal to or greater than 210 $\frac{\mathrm{lm}}{W}$, the light evaluated is in class A.

Labeling the energy efficiency of an IoT device with energy harvesting is more complicated because the parameters that frame the value on the scale should vary based on the irradiance level and the number of hours of light in the place where the devices are installed. Otherwise, all the IoT solutions deployed in northern European countries would have a lower energy class than other European countries because they need larger components. Furthermore, for very sensitive applications, the oversizing of the components is justified within certain limits. Moreover, to define the values of the multiplying factor ($F_{TM}$), more than four scenarios must be considered, for example, the different fields of application and if the device is connected to the mains, and it should be considered that an IoT device is often built ad hoc for a specific solution. Therefore, it is difficult (if ever possible) to compare devices made up of different elements but that have the same purpose. The label should not only consider the energy efficiency but also the waste impact of these devices, so it should be necessary to define a way to combine the two impacts.

## Figures and Tables

**Figure 1 sensors-23-01537-f001:**
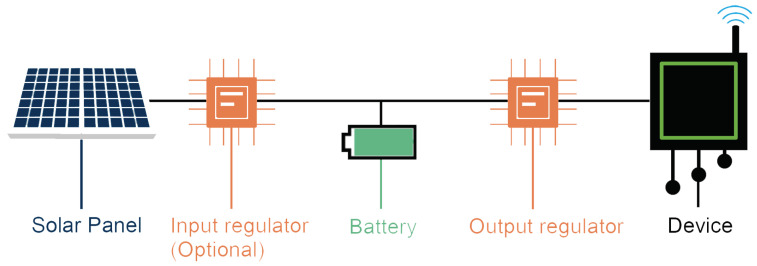
System components.

**Figure 2 sensors-23-01537-f002:**
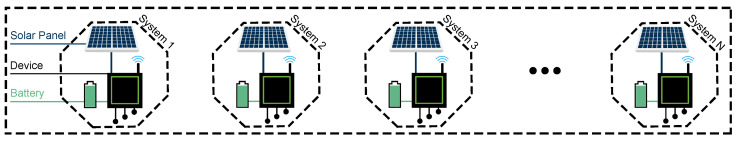
IoT sense entities.

**Figure 3 sensors-23-01537-f003:**
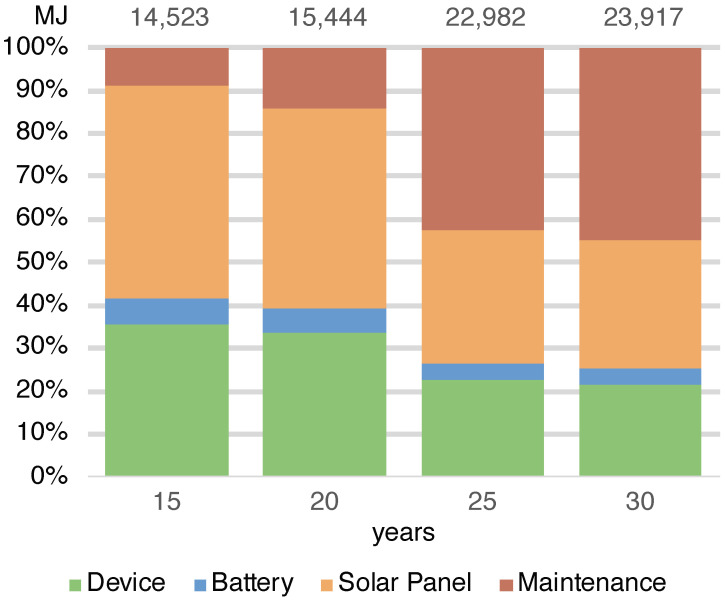
Energy impact of the IoT solution considering 35 devices in different lifespan scenarios.

**Figure 4 sensors-23-01537-f004:**
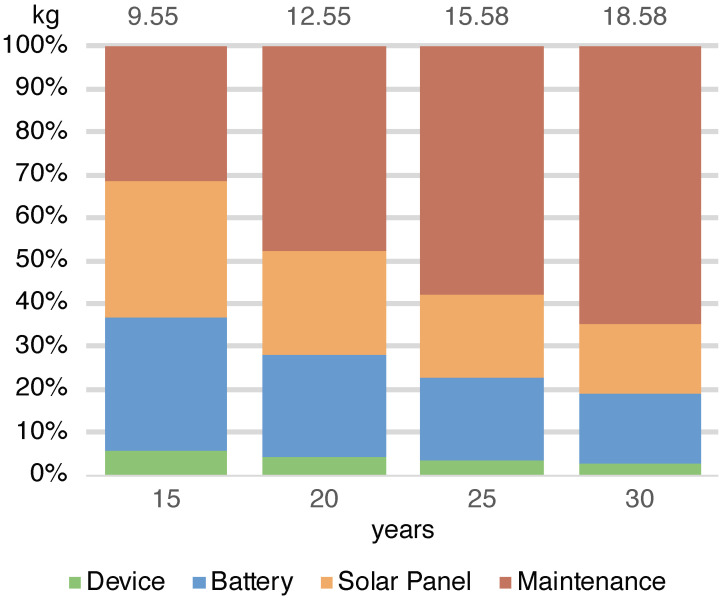
Waste produced by IoT solution considering 35 devices in different lifespan scenarios.

**Figure 5 sensors-23-01537-f005:**
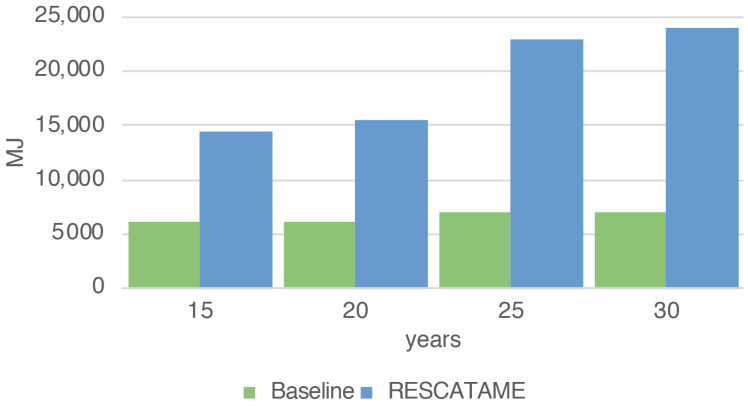
Energy impact comparison of the two solutions in four lifespan scenarios.

**Figure 6 sensors-23-01537-f006:**
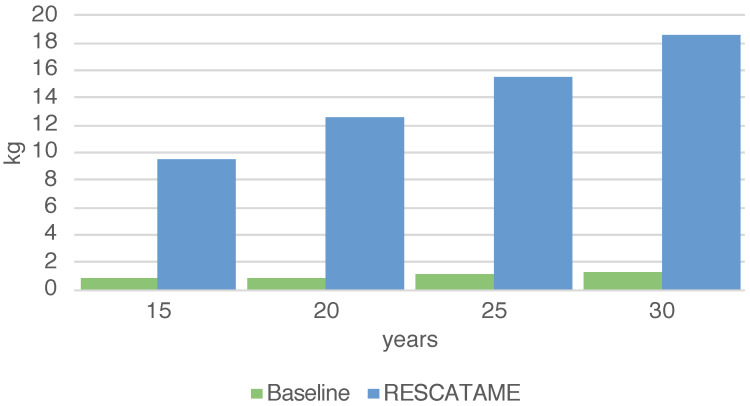
Waste impact comparison of the two solutions in four lifespan scenarios.

**Figure 7 sensors-23-01537-f007:**
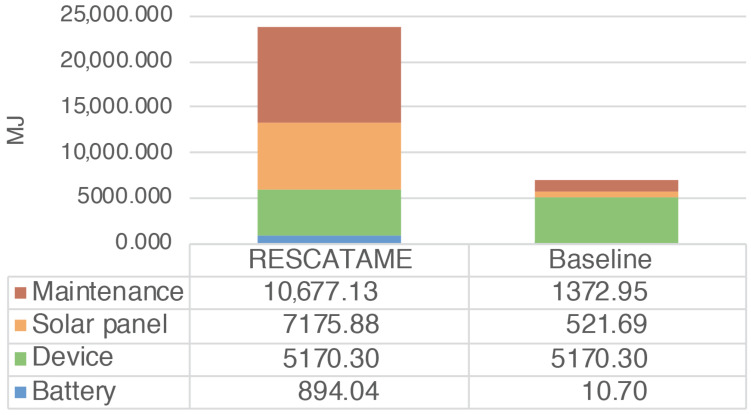
Energy impact of RESCATAME project and baseline green solution.

**Figure 8 sensors-23-01537-f008:**
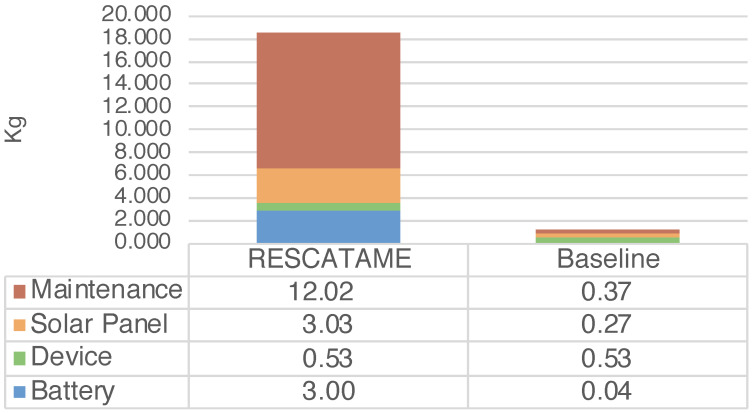
Waste produced by RESCATAME project and baseline green solution.

**Figure 9 sensors-23-01537-f009:**
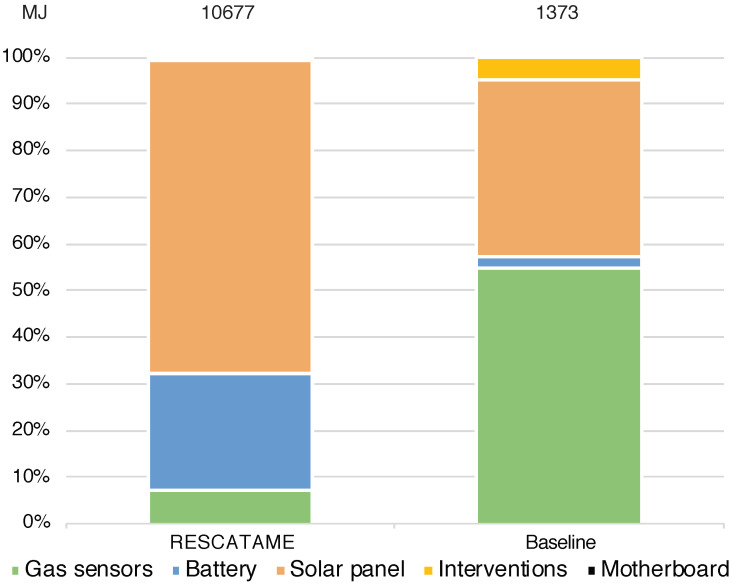
Energy impact of the maintenance of RESCATAME project and baseline green solution.

**Figure 10 sensors-23-01537-f010:**
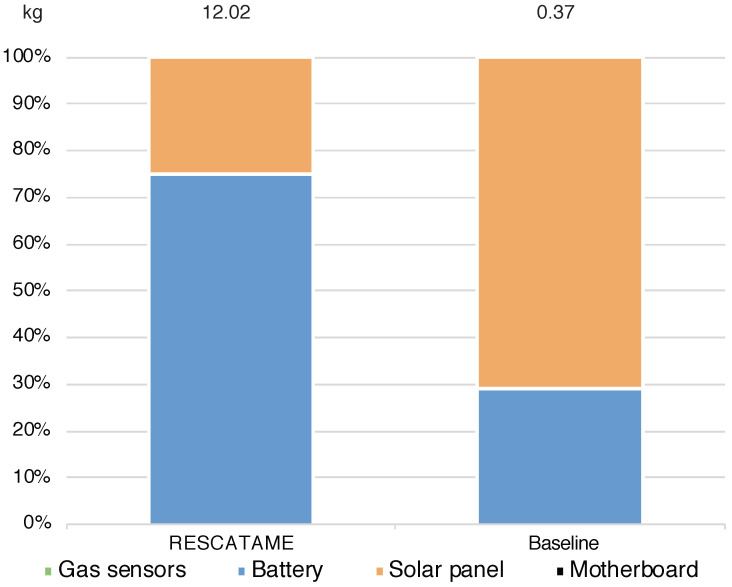
Waste impact of the maintenance of the RESCATAME project and baseline green solution.

**Figure 11 sensors-23-01537-f011:**
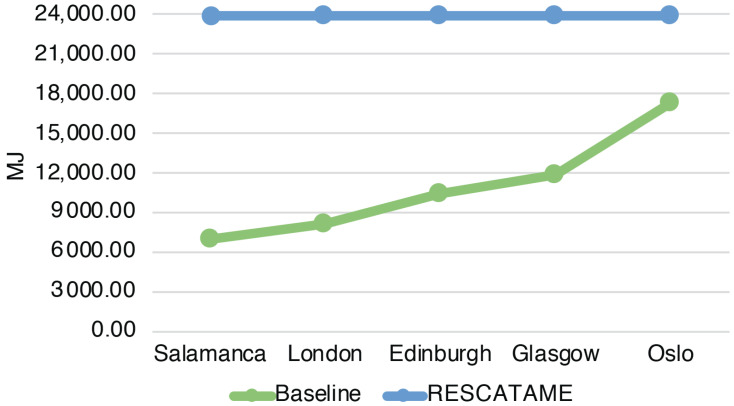
Energy impact comparison between the RESCATAME implementation and the green solution in the different cities of installation.

**Figure 12 sensors-23-01537-f012:**
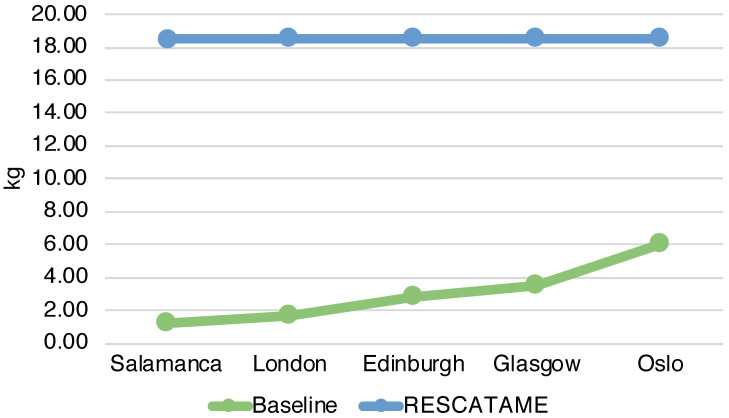
Waste impact comparison between the RESCATAME implementation and the green solution in the different cities of installation.

**Figure 13 sensors-23-01537-f013:**
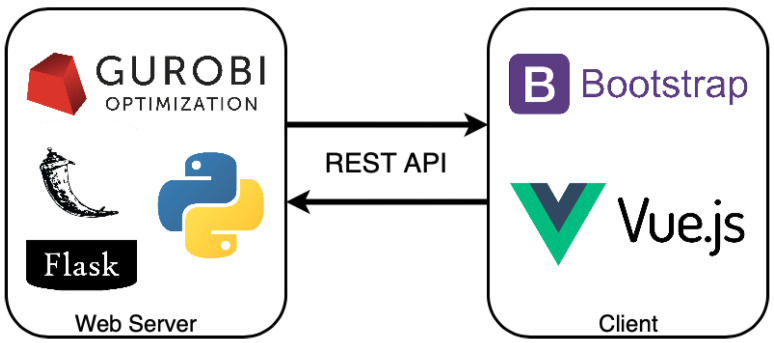
Web service architecture.

**Figure 14 sensors-23-01537-f014:**
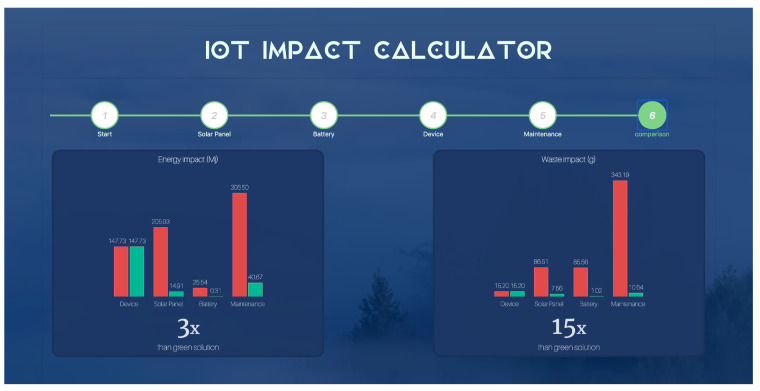
Web service user interface.

**Table 1 sensors-23-01537-t001:** Summary of key differences with closest related work.

Reference	Devices Considered	Energy Harvesting	Life-Cycle Stage	Definition of a Model/Algorithm
[1]	Desktops, smartphones, tablets, datacenters and communication networks	-	Production and operational life stages	Model
[6]	Datacenters and fog devices	-	Operational life stage	Model
[7]	IoT and fog devices	-	Operational life stages	Model
[35]	Smartphones	-	Full life-cycle assessment	Model
[36]	Smartphones	-	Full life-cycle assessment	Model
[37]	IoT devices	-	Operational life stage	Algorithm
[38]	IoT devices	-	Operational life stage	Algorithm
[39]	IoT medical devices	-	Operational life stage	Algorithm
Our work	IoT devices	Solar energy harvesting system	Full life-cycle assessment	Model

**Table 2 sensors-23-01537-t002:** Datasheet parameters of the device elements.

Element	Product	Area [cm${}^{2}$]	On [mA]	Sleep [mA]	Ref.
MCU	ATmega-12	2.64	9	0.06	[47]
Radio	XBee-802.15.4	6.73	0.26	0	[47,48]
CO	TGS-2442	0.67	3	0	[49,50]
NO_2_	MICS-2710	0.64	26	0	[50,51]
O_3_	MICS-2610	0.64	34	0	[50,51]
Particles	GP2Y1010AU0F	13.8	11	0	[52,53]
Noise	WM-61B	0.28	0.5	0	[53,54]
Temp.	MCP9700A	0.28	0.01	0	[53,55]
Humidity	808H5V5	0.98	0.38	0	[53,56]

**Table 3 sensors-23-01537-t003:** Impact of energy and waste of one IoT device with energy harvesting in the three phases of the life-cycle.

Phase	Device	Solar Panel	Battery
Manufacturing (MJ)	147.72	205.03	25.54
Utilization (MJ)	$1.22\times 10^{-3}$	0.03	-
Disposal (kg)	0.02	0.09	0.09

**Table 4 sensors-23-01537-t004:** Total impact of energy and waste caused by the maintenance of one system according to four different application lifespan.

Maintenance Impact		
Years	MJ	kg
15	36.96	0.09
20	63.85	0.17
25	278.94	0.26
30	306.51	0.34

**Table 5 sensors-23-01537-t005:** Parameters needed to size solar panel and battery where ρ is the solar radiation [42], $\epsilon (p_{i})$ is the solar panel efficiency [64], $wo(p_{i})$ is the efficiency of the solar panel in the 20th year [19], $E(d_{i})$ is the device energy requirement, $\epsilon (b_{i})$ is the battery efficiency [20], $\epsilon (r_{i})$ is the efficiency of the output regulator [57] and δ is the percentage of hours of light [65].

ρ	$\epsilon (p_{i})$	$\mathrm{wo}(p_{i})$	$E(d_{i})$	$\epsilon (b_{i})$	$\epsilon (r_{i})$	δ
4.06 MJ	17%	80%	$1.22\times 10^{-3}$ MJ	90%	90%	37.5%

**Table 6 sensors-23-01537-t006:** Energy impact ratio between RESCATAME and baseline solution.

years	15	20	25	30
$\nu_{\mathrm{energy}}$	2.4	2.5	3.3	3.4

**Table 7 sensors-23-01537-t007:** Waste impact ratio between RESCATAME and baseline solution.

years	15	20	25	30
$\nu_{\mathrm{waste}}$	11	13.9	13.3	15.4

**Table 8 sensors-23-01537-t008:** Solar panel size and battery capacity to satisfy the daily device load based on the irradiation level and the hours of light in the city of installation.

Parameter	Salamanca	London	Edinburgh	Glasgow	Oslo
irradiation $\left({\mathrm{MJ}/m}^{2}\right)$	4.06	1.97	0.97	0.72	0.38
hours of light	9	7	6	6	5
$size(p_{i})$ $\left(m^{2}\right)$	$2.72\times 10^{-3}$	$5.6\times 10^{-3}$	0.01	0.02	0.03
$capacity(b_{i})$ (MJ)	$0.94\times 10^{-3}$	$1.06\times 10^{-3}$	$1.13\times 10^{-3}$	$1.13\times 10^{-3}$	$1.19\times 10^{-3}$

**Table 9 sensors-23-01537-t009:** Energy impact ratio between RESCATAME and baseline solution.

City	Salamanca	London	Edinburgh	Glasgow	Oslo
$\nu_{energy}$	3.4	2.9	2.3	2	1.4

**Table 10 sensors-23-01537-t010:** Waste impact ratio between RESCATAME and baseline solution.

City	Salamanca	London	Edinburgh	Glasgow	Oslo
$\nu_{waste}$	15.4	10.9	6.7	5.4	3

## Data Availability

Not applicable.

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
