# Peer review of "Estimating the Environmental Impact of Green IoT Deployments"

_sensors, 2023, doi:10.3390/s23031537_

Round 1
Reviewer 1 Report
The subject is very relevant. However:
1. Authors mix IoT and ICT definitions inappropriately.
2. Thus, the title of the work must be revised.
3. Between items 2. and 2.2 there should be a small introduction.
4. Why was software not used for ACL calculations?
5. How did you arrive at eq. (1)? Why not include other terms?
6. The presentation of the "case study" is very confusing.
7. Between items 8. and 8.1 there should be a short introduction.
8. The way the conclusions are presented is not the usual one, why was it done this way?
9. Much of the content of the conclusions is not supported by the research carried out, being, therefore, perhaps, discussions.
10. There is no clear connection between current work and "future work".
Author Response
Responses to the reviewer comments
We thank the reviewer for the work of revision of our manuscript. We have worked to fix all the issues highlighted by the reviewer as follows:
- Authors mix IoT and ICT definitions inappropriately.
We agree on the fact that we used the term ICT in many cases in an inappropriate way. The manuscript has been rewritten to avoid the use of the term ICT. In addition to this, we noticed we did not define the IoT acronym at the beginning of the introduction. We also fixed this.
- Thus, the title of the work must be revised.
As in response to point 1 we avoided the inappropriate and excessive use of the term ICT, and clarified that the work focuses on IoT, we would keep the current title of the paper.
- Between items 2. and 2.2 there should be a small introduction.
We have added the following introduction to section 2:
“This section presents the background information on which the rest of the work leverages. Specifically, we introduce information about the impact of production and disposal of the different components of a solar energy harvesting IoT device, namely the device itself, the solar panel and the batteries (sections 2.1, 2. and 2.3). We also introduce the concept of predictive maintenance in Section 2.4 and, finally, we present a general architecture of energy harvesting IoT devices in Section 2.5.”
- Why was software not used for ACL calculations?
We are not sure we correctly interpreted this question (the term ACL does not appear in the text)… our interpretation is that this question refers to the application of our model to the case study.
The fact is that part of the calculations in the case study requires a lot of input data and just calculation of equations for which it is easier to proceed by using solvers connected to spreadsheet data. However, there are other parts of the calculus which instead required software, in particular that to assess the cost of maintenance, and for which we used software (in python, based on the Gurobi Python library) to solve the corresponding optimization problem.
Probably this was unnoticed by the reviewer because the note on this software was written in another section.
For this reason, to clarify that this calculation had been achieved by means of a software in python using the Gurobi library, we rephrased the following statement in Section 6.1.3:
“Through the multi-objective linear programming problem, we find the optimal scheduling strategies to minimize the energy and waste impacts considering 15, 20, 25 and 30 years as application lifespan scenarios.”
As follows:
“Through the multi-objective linear programming problem, and by means of a python solver based on the Gurobi Python library, we find the optimal scheduling strategies to minimize the energy and waste impacts considering 15, 20, 25 and 30 years as application lifespan scenarios.”
- How did you arrive at eq. (1)? Why not include other terms?
We defined E_s and the corresponding Equation 1 as the energy costs for the production and operational life of a deployment comprising m IoT devices. The equation sums the individual contribution of each single device in terms of the energy required to produce and keep operational all of its components. We agree on the fact that this model abstracts energy costs of the deployment that may depend on the environment and location where the devices are deployed. This is a limitation of the work, and we therefore include this consideration in Section 8.2 “Limitations”, by rewriting the statement
“Finally, we disregard aspects like recycling, the impact of environmental and operative conditions on the lifetime of the devices and components.”
As follows:
“Finally, we disregard aspects like recycling, the impact of environmental and operative conditions on the lifetime of the devices and components and costs of deployment that may depend on the location or the environment of the deployment itself.”
- The presentation of the "case study" is very confusing.
Probably the problem comes from the fact that we did not provide at the beginning of the section an appropriate rationale and structure of the case study.
We have changed (and extended) the introductory phrase of the section as follows:
“We now illustrate the effectiveness of our model by analysing its application to a real IoT deployment. To this purpose we selected the IoT deployment implemented in the RESCATAME European project which includes a significant number of solar-power, energy harvesting IoT devices deployed in the Spanish city of Salamanca. This analysis is organized in four parts. In the first part (Section 6.1), after revising briefly the main aspect of the project we assess the impact of its devices, energy harvesting subsystem and maintenance, and we conclude this section by presenting the aggregate results of this analysis. In the second part (Section 6.2) we present a baseline green solution which is functionally equivalent to that used for RESCATAME, but with a smaller impact, and in the third part (Section 6.3) we compare the energy and waste impact of RESCTAME and of its baseline green solution.
The last part (Section 6.4), still referring to the RESCATAME deployment, discusses the effect of different geographical locations at different latitudes in terms of energy and waste impact.”
- Between items 8. and 8.1 there should be a short introduction.
We have added the following introduction to section 8:
“As concluding remarks we summarize the contribution of the work (Section 8.1), we thoroughly discuss the limitations of our approach (Section 8.2), and, finally, we present our ongoing and future work on this research area (Section 8.3).”
- The way the conclusions are presented is not the usual one, why was it done this way?
Yes, that’s true, considering the complexity of the problem and the consequent number of limitations that are present in our work (and of which we are aware), we preferred to organize this section in three parts so to better emphasize the limitations, by giving them a subsection on their own.
We hope the reviewer accepts this approach and that we can keep the current structure of the conclusions. On the other hand, we fixed the problems in this section highlighted by the reviewer in comments 7, 9 and 10.
- Much of the content of the conclusions is not supported by the research carried out, being, therefore, perhaps, discussions.
We agree, we have changed the title of Section 8 in “Discussion and conclusions”.
- There is no clear connection between current work and "future work".
The reviewer is right on this point, as we are already undertaking the future work. We have changed the title of the section in “Current and future work”
Reviewer 2 Report
The manuscript proposes an analytical methodology to assess the environmental impact of IoT deployments. The following points are observed.
1. The manuscript discusses a very critical issue which is highly relevant in the current scenarios.
2. The manuscript organization is missing in the introduction section. It is suggested to provide it for better readability and manuscript flow.
3. A tabular comparison of the existing approaches with the proposed approach can provide a better understanding and improvements proposed, which I found is missing.
Author Response
Responses to the reviewer comments
We thank the reviewer for the work of revision of our manuscript. We have worked to fix all the issues highlighted by the reviewer as follows:
- The manuscript discusses a very critical issue which is highly relevant in the current scenarios.
We thank the reviewer for appreciating our work.
- The manuscript organization is missing in the introduction section. It is suggested to provide it for better readability and manuscript flow.
We have added the following sentence at the end of the introduction:
“The rest of the work is organized as follows: Section 2 presents the background of our work, including the information about the impact of production and disposal of IoT devices, and the methods to assess the costs for the maintenance of an IoT deployment, and Section 3 reviews the state of the art. We then introduce in Section 4 the model of IoT deployments to which we refer in the rest of the work, in Section 5 we introduce our model to estimate the impact of IoT deployments, and in Section 6 we present a case study based on the deployment of the RESCATAME European project. In Section 7 we introduce the tool to assess the impact of IoT deployments and, finally, in Section 8 we summarize the main contributions of the work, we analyze its limitations, and we discuss the future work.”
- A tabular comparison of the existing approaches with the proposed approach can provide a better understanding and improvements proposed, which I found is missing.
As suggested by the reviewer we have inserted a table at the end of section 3 (State of the art) which compares the different approaches.
Round 2
Reviewer 1 Report
The authors have sufficiently improved the paper.